# Genetic Diversity and Expression of Intimin in *Escherichia albertii* Isolated from Humans, Animals, and Food

**DOI:** 10.3390/microorganisms11122843

**Published:** 2023-11-23

**Authors:** Qian Liu, Xi Yang, Hui Sun, Hua Wang, Xinxia Sui, Peihua Zhang, Xiangning Bai, Yanwen Xiong

**Affiliations:** 1National Key Laboratory of Intelligent Tracking and Forecasting for Infectious Diseases, National Institute for Communicable Disease Control and Prevention, Chinese Center for Disease Control and Prevention, Beijing 102206, China; liuqian_icdc@163.com (Q.L.); yangxi@icdc.cn (X.Y.); sunhui@icdc.cn (H.S.); wanghuamed@163.com (H.W.); 15588519633@163.com (X.S.); zhangphzzu@163.com (P.Z.); xiangning.bai@ki.se (X.B.); 2Division of Laboratory Medicine, Oslo University Hospital, 0372 Oslo, Norway

**Keywords:** *Escherichia albertii*, intimin, locus of enterocyte effacement, attaching and effacing lesion

## Abstract

*Escherichia albertii* (*E. albertii*) is an emerging diarrheagenic pathogen associated with sporadic infections and human gastroenteric outbreaks. The *eae* gene, which encodes intimin in the locus of enterocyte effacement (LEE) operon, contributes to the establishment of the attaching and effacing (A/E) lesion. Increasing collection of *E. albertii* strains from various sources has resulted in a rapid increase in the number of *eae* subtypes. This study systematically investigated the prevalence and genetic diversity of *eae* among *E. albertii* strains isolated from humans, animals, and food. The *eae* gene was present in 452/459 (98.5%) strains and 23 subtypes were identified including two novel subtypes, named *eae*-α11 and η3. The *eae*-σ subtype was the most predominant among humans, animals, and food-derived strains, while *eae*-γ3, τ, and α11 were unique in human-derived strains. Additionally, the LEE island was also analyzed at genomic, transcriptional, and functional levels through genomic analysis, quantitative reverse transcription PCR, and HEp-2 cell adherence assays, respectively. The *eae* transcript levels were variable and associated with *eae* subtypes. Three different adherence patterns, including localized adherence-like (LAL), diffuse adherence (DA), and detachment (DE), were observed among *E. albertii* strains. This study demonstrated a high diversity of functional intimin in *E. albertii* strains isolated from humans, animals, and food. Further in vivo and in vitro studies are warranted to better elucidate the role of intimin or LEE in different genetic backgrounds.

## 1. Introduction

*Escherichia albertii* (*E. albertii*), a Gram-negative foodborne gastrointestinal pathogen, is the newest member of the attaching and effacing (A/E) morphotype of bacterial pathogens [1]. This lesion is characterized by the intimate adherence between the bacteria and the host cell, leading to the destruction of microvilli and the formation of pedestal-like structures beneath the adherent bacteria [2]. The genetic basis for the formation of the A/E lesion is a chromosomal pathogenicity island named locus of enterocyte effacement (LEE). The LEE is a ~35 kb genetic element that encodes a type III secretion system (T3SS), the outer membrane protein intimin and its translocated receptor Tir, as well as some secreted effectors that are linked to virulence [3]. In addition to LEE, cytolethal distending toxin (CDT) genes and Shiga toxin 2 genes (*stx2a* and *stx2f*) also contribute to the pathogenesis of *E. albertii* [4]. The *cdt* operon contains three adjacent genes, *cdtA*, *cdtB,* and *cdtC*. CdtB acts as an active subunit with DNase I activity, CdtA and CdtC facilitate the binding of CDT to receptor molecules on susceptible cells and entry of CdtB into the cytoplasm [5]. Currently, the *cdtB* gene has been divided into six subtypes (*cdtB*-I to *cdtB*-VI) in *E. albertii* strains [6]. The *stx* gene encoding for Shiga toxins has been found in certain strains of *E. albertii*, which is closely related to clinically significant *E. albertii* infection [7,8].

*E. albertii* was reported to be responsible for six human outbreaks in Japan from 2003 to 2015 [9]. Clinical symptoms caused by *E. albertii* infection are similar to those caused by enteropathogenic *E. coli* (EPEC), typically including watery diarrhea, dehydration, abdominal pain, vomiting, and fever [9]. In 2004, large-scale mortality of finch species (*Carduelis flammea*) occurred in Alaska, and *E. albertii* was identified as the probable etiology [10]. In recent years, *E. albertii* strains have been identified widely in avian, mammal species, raw meats, and humans [11,12,13,14]. However, the close association of animal or food vehicles with human infections remains unclear.

The intimin encoded by *eae* mediates bacterial attachment to epithelial cells [3]. The entire length of the *eae* gene is approximately 2800 nucleotides. The *eae* gene has been divided into several subtypes based on the diversity of the 3′ region, which has been identified to be the intimin cell-binding domain (Int280a) [15]. Recent surveys of *E. albertii* strains from various sources have identified several *eae* subtypes based on sequence variations [6,16]. Importantly, some of the intimin subtypes identified in *E. albertii* strains have been novel or rare subtypes in *E. coli*. Further study is required to better understand the diversity and functions of intimin subtypes in *E. albertii*. In this study, we investigated the prevalence and genetic diversity of the intimin gene among *E. albertii* strains isolated from diverse sources, and analyzed LEE island at both genomic and functional levels.

## 2. Materials and Methods

### 2.1. Isolates Collection

A total of 205 whole genome sequenced *E. albertii* strains from China were used in this study, including 201 strains previously reported [6,12], one newly sequenced strain ESA302 in this study, and three collected from the NCBI database (https://www.ncbi.nlm.nih.gov/datasets/, accessed on 9 October 2022). The genomic DNA of strain ESA302 was extracted using the Wizard Genomic DNA purification kit (Promega, WI, USA). Fragment libraries of the genomic DNA were generated using the Universal DNAseq Library Prep Kit (KAITAI-BIO, Hangzhou, China) and sequenced using the combined methods of the PacBio Sequel (Pacific Biosciences, Menlo Park, CA, USA) and Illumina NovaSeq 6000 platform (Illumina, San Diego, CA, USA). After filtering out low-quality reads, the clean data were de novo assembled into a complete genome using Unicycler v0.4.8 [17]. In addition, the whole genome sequences of 254 *E. albertii* strains from humans (*n* = 139), animals (*n* = 114), and food (*n* = 1) isolated in different countries from 1954 to 2022 were downloaded from the NCBI database. QUAST v5.2.0 was used to assess the quality of genomes [18]. One reported specific gene (EAKF1_ch4033) of *E. albertii* was used to confirm the species level of all genomes used in this study, with 70% coverage and 90% identity [6,19]. All 459 *E. albertii* genomes used in this study are listed in Appendix A.

### 2.2. E. albertii O-Antigen Genotyping, stx2, and cdtB Subtyping

The *E. albertii* O-antigen genotypes (EAOgs) were performed by BLASTN (https://ftp.ncbi.nlm.nih.gov/blast/executables/blast+/LATEST/, accessed on 3 January 2023) search using the nucleotide sequences of the 42 primer pairs described by Ooka et al. [20]. Only 100% matching was assigned to a given genotype. To predict the *cdtB* and *stx2a*/*stx2f* genes, an in-house subtyping database was created with the ABRicate 1.0.1 (https://github.com/tseemann/abricate, accessed on 3 January 2023) by including representative nucleotide sequences of all identified subtypes. The assemblies were then searched against the subtyping database. The reference sequences for *cdtB* and *stx2* genes (*stx2a* and *stx2f*) were summarized in Appendix A.

### 2.3. eae Subtyping and Polymorphisms Analysis

To predict the intimin subtypes of each *E. albertii* strain, representative nucleotide sequences were downloaded and organized from GenBank according to Ooka et al. and Luo et al. [6,16]. The BLASTN search with an identity of ≥95% and coverage of ≥70% was used to determine the intimin subtypes. A 95% nucleotide sequence identity cutoff value and phylogenetic tree structure were used to define a novel subtype as described previously [15]. Then, the complete *eae* sequences of all *E. albertii* isolates were extracted from assembles, and aligned using the MAFFT program (https://mafft.cbrc.jp/alignment/software/, accessed on 10 January 2023) [21]. The *eae* genotype (GT) based on *eae* sequence polymorphism was used to determine the diversity within each *eae* subtype [15].

### 2.4. Locus of Enterocyte Effacement (LEE) Analyses

The nucleotide sequences of LEE elements were manually located through annotations and then extracted by software UGENE version 46 [22] from all complete genomes. The genetic structure of LEE elements was visualized by Easyfig version 2.2.2 [23]. Alignments of the nucleotide sequences of LEE were created using MAFFT [21], and neighbor-joining trees were constructed using MEGA 11 [24] with default settings and visualized using phytools v1.0 (https://github.com/liamrevell/phytools.git, accessed on 11 January 2023) [25].

### 2.5. Pangenomes Analysis

Genome assemblies were annotated using Prokka v1.14.6 [26], and pangenomes of *E. albertii* strains were then calculated from genome annotations using Roary version 3.13.0 (https://github.com/sanger-pathogens/Roary, accessed on 5 December 2022) [27] with the command: roary -s -e -mafft *.gff. Pangenomes consist of a complete set of core and accessory genes in all analyzed isolates.

### 2.6. mRNA Expression Level of LEE-Related Genes

Bacteria were grown in 5 mL of Luria–Bertani (LB) with shaking (180 rpm) at 37 °C to reach an optical density at 600 nm (OD_600_) of 0.5. Total RNA was extracted from the bacterial cultures using RNeasy Mini Kit (Qiagen, Valencia, CA, USA) according to the manufacturer’s instructions. Next, the total RNA amount was determined by a NanoDrop 2000 spectrophotometer (Thermo Fisher Scientific, Waltham, MA, USA). Quantitative reverse transcription PCR (RT-qPCR) was accomplished using HiScript II One Step qRT-PCR SYBR Green Kit (Vazyme, Nanjing, China) with *eae*, *ler* primers (0.4 μM) and *gapA* universal primers (0.4 μM), respectively (Appendix A). The RT-qPCR cycle parameters were as follows: 50 °C (15 min), 95 °C (30 s), 40 cycles of 95 °C (10 s), 55 °C (30 sec-read fluorescence), and followed by melt curve analysis. Each experiment was calculated with three technical replicates. The relative difference in gene expression was calculated using the 2^−ΔΔCT^ method [28]. The *E. albertii* type strain NBRC 107761, 27 *E. albertii* isolates (10 strains from animals, 9 strains from food, 8 strains from humans), typical EPEC strain E2348/69, and three atypical EPEC isolates were tested. The bar plots were visualized using the web application Chiplot [29].

### 2.7. Cell Adherence Assays

The HEp-2 adherence patterns of *E. albertii* strains were determined according to the method described by Cravioto et al. [30]. Briefly, bacteria strains were grown in LB at 37 °C to OD_600_ = 0.5. HEp-2 cells cultivated for 48 h in 24-well plates containing coverslips were infected with bacterial strains at a multiplicity of infection (MOI) of 1:100. After 3 h of incubation at 37 °C, preparations were washed with phosphate-buffered saline (PBS), fixed with methanol, stained with Giemsa stain, and examined by light microscopy. When the adherence pattern was weak or negative, a new preparation was made and examined after a 6 h incubation period. Light microscopy was used to classify adherence patterns as previously described: localized adherence (LA)—large, compact microcolonies visualized after 3 h of interaction; localized adherence-like (LAL)—looser bacteria clusters than LA and identified after 6 h of interaction; aggregative adherence (AA)—a “stacked-brick” arrangement; diffuse adherence (DA)—bacteria attached in a randomly scattered manner; nonadherent (NA)—without strains adherent to cell; detachment (DE)—cell detached from the dishes [31]. A total of 32 strains were selected in this section, including 28 *E. albertii* strains and four *E. coli* strains that served as different adhesion types: tEPEC E2348/69 for LA, aEPEC 019 for LAL, EAEC 042 for AA, and *E. coli* HB101 for NA.

### 2.8. Statistical Analyses

Fisher’s exact test was used to analyze the association between *eae* subtypes and their distribution in different sources. The statistical significance was determined by SPSS Statistics26, and *p* value < 0.05 was considered statistically significant.

## 3. Results

### 3.1. Prevalence of eae, cdtB, and stx2f Genes in E. albertii Strains

Among the contained *E. albertii* strains, 452/459 (98.5%) strains were positive for the *eae* gene, and 23 subtypes were detected (Table 1). The *cdtB* gene was present in 455/459 (99.1%) strains, belonging to four *cdtB* subtypes. The predominant *cdtB* subtypes were *cdtB*-II and *cdtB*-VI, accounting for 60.2% (274/455) and 24.8% (113/455), respectively. Notably, there were 47/455 (10.3%) strains possessing two *cdtB* subtypes each, e.g., *cdtB*-I/II, *cdtB*-I/VI, or *cdtB*-II/IV. None carried *cdtB*-III or *cdtB*-V subtype. Additionally, the *stx2f* gene was detected in 52/459 (11.3%) strains, while *stx2a* was absent in all strains (Appendix A).

### 3.2. Prevalence of E. albertii O-Antigen Genotypes

Among 459 strains, 422 strains were classified into 40 different *E. albertii* O-antigen genotypes, and 37 strains were untypable. The most predominant *E. albertii* O-antigen genotype was EAOg4 (101/459, 22.0%), followed by EAOg1 (73/459, 15.9%) and EAOg2 (27/459, 5.9%) (Appendix A).

### 3.3. Diversity and Subtypes of eae in E. albertii Strains from Different Sources

A total of 23 *eae* subtypes were identified, with *eae*-σ (*n* = 186), *ρ* (*n* = 47), and *ε3* (*n* = 29) being the dominant subtypes (Table 1). The *eae*-σ subtype was predominant among human-, animal-, and food-derived strains, accounting for 29.2% (43/147), 28.0% (42/150), and 62.3% (101/162), respectively. The *eae*-α8, α10, β3, β4, ε1, ε3, ε4, λ2, σ, σ2, ι2, ο1, ν, and τ subtypes were associated with *cdtB*-II subtype with a significant difference (*p* < 0.01). In addition, two novel *eae* subtypes named *eae*-α11 and η3 were defined based on the sequence similarity and phylogenetic relationship (Appendix A). The *eae*-α11 and η3 were presented in strains derived from humans and animals, respectively.

In human-derived strains, 22 *eae* subtypes were detected. The predominant *eae* subtypes were *eae*-σ (43/147, 29.2%) and ι2 (15/146, 10.3%). The *eae*-ι2 was mainly present in human-derived strains, accounting for 83.3% (15/18). In addition, *eae*-γ3 (*n* = 3), τ (*n* = 6), and α11 (*n* = 4) were only present in human-derived strains (Table 1).

In animal-derived strains, 19 *eae* subtypes were detected. The predominant *eae* subtypes were *eae*-σ (42/150, 28.0%), α9 (20/150, 13.3%), and σ2 (16/150, 10.7%). *eae*-α9, ε4, and σ2 were mainly present in animal-derived strains (*p* < 0.001), accounting for 80.0% (20/25), 92.3% (12/13), and 80.0% (16/20), respectively (Table 1).

In food-derived strains, 7 *eae* subtypes were detected. The predominant subtypes were *eae*-σ (101/162, 62.3%), ρ (36/162, 22.2%), and ε3 (15/162, 9.3%). The *eae*-ρ was mainly present in food-derived strains, accounting for 76.6% (36/47). The prevalence of *eae*-ρ in food was significantly higher than in humans or animals (*p* < 0.001) (Table 1).

### 3.4. Genotypes of eae Subtype and Its Correlation with Sources

The predominant *eae* subtypes σ, ρ, ε3, and α9 were also analyzed to determine the diversity within each *eae* subtype (Appendix A). The reference sequence of each subtype was assigned as genotype 1 (GT1) and summarized in Appendix A.

Among the 186 *eae*-σ strains, 8 genotypes (σ/GT2-GT9) were identified using *eae*-σ sequence AJ781125 as reference. Compared to σ/GT1, σ/GT2 had a synonymous substitution at location 487 (T to C) and other genotypes were non-synonymous (Figure 1A). The σ/GT2 was the major genotype in *eae*-σ strains and was associated with food-derived strains (Figure 1B).

Among the 47 *eae*-ρ strains, 5 genotypes (ρ/GT2-GT6) were defined based on the *eae*-ρ reference sequence (DQ523613). The mutations ρ/GT2-GT6 were all non-synonymous, with each of them displaying two non-synonymous substitutions at locations 2188 (A to T) and 2242 (C to A). These substitutions resulted in the change from isoleucine to leucine and from glutamine to lysine, respectively. ρ/GT4 was the predominant genotype in *eae*-ρ strains and was associated with animal-derived strains (Appendix A).

Among the 29 *eae*-ε3 strains, 7 genotypes (ε3/GT2-GT8) were defined based on the *eae*-ε3 reference sequence (AJ7876649). All variants of *eae*-ε3 were non-synonymous substitutions. The major genotype was ε3/GT2, which was associated with food-derived strains (Appendix A).

Among the 25 *eae*-α9 strains, 8 genotypes (α9/GT2-GT9) were defined based on the *eae*-α9 reference sequence (GCA_003860365.1). Except for α9/GT3 and GT5, other variants of *eae*-α9 were non-synonymous. α9/GT3 had only one synonymous substitution at location 489 (A to C), while α9/GT5 had two synonymous substitutions at locations 2466 (C to A) and 2733 (T to C). The major genotype was α9/GT6, which was associated with animal-derived strains (Appendix A).

### 3.5. The Locus of Enterocyte Effacement in E. albertii

The complete genomes of 40 *E. albertii* strains were selected to characterize the LEE elements. The LEE elements were composed of 41 open reading frames organized in six major operons. Meta-alignment revealed that LEE elements were conserved between *E. albertii* and the other A/E members (Appendix A). In all 40 *E. albertii* strains, the LEE elements were integrated into the tRNA-*pheU* loci and had a length of 34–35 kb, with 81–100% similarity among each other. A neighbor-joining phylogenetic tree based on the complete nucleic acid sequences of the LEE elements was then constructed. According to the topological structure and evolutionary distance, the phylogenetic tree based on the sequences of LEE elements was divided into three main clades (L-Clade 1, 2, and 3) (Figure 2A). Strains with the same *eae* subtypes clustered together. L-Clades 1 and 3 contained strains from humans, animals, and food, while Clade 2 only harbored strains from humans. Within L-Clade 1, the predominant *eae* subtypes were *eae*-ο, ο1, ι2, ν, τ, and ρ. L-Clade 2 mainly consisted of *eae*-α8, α9, and σ2 subtypes. L-Clade 3 predominantly possessed *eae*-α10, ε1, ε3, ε4, υ, β4, ξ, and σ subtypes.

To explore concordance between LEE phylogenetic and pan-genome tree, a pan-genome tree was constructed using the 40 complete genomes of *E. alberti*, which was compared to that of LEE. The pan-genome tree also formed three major clades, namely G-Clade I, II, and III. Each clade contained strains derived from humans, animals, and food. The result indicated a significant divergence relationship for LEE, which was reflected in the tanglegrams (Figure 2). Only L-Clade 1 of the LEE phylogenetic tree was correlated with G-Clade II of the pangenome tree. L-Clade 2 of the LEE phylogenetic tree was correlated with a portion of G-Clade I of the pangenome tree. L-Clade 3 of the LEE phylogenetic tree was correlated with both G-Clades I and III of the pangenome tree.

### 3.6. LEE Genes Expression in Different Strains

The *ler* gene, encoded in the LEE1 operon, is the master transcriptional factor of the six LEE operons [3]. The *eae* gene, which encodes intimin in the LEE5 operon, contributes to the establishment of the attaching and effacing (A/E) lesion [1]. The expression levels of *ler* and *eae* genes were evaluated using qRT-PCR assays. Compared to strain NBRC 107761, all *E. albertii* isolates presented *ler* and *eae* transcript levels varying from 0.5 to 4.2-fold and 0.2 to 4.4-fold, respectively. Strain ESA298 isolated from animal source with *eae*-ε3 subtypes displayed the highest transcript level of *ler* gene, while its transcript level of *eae* gene was relatively low. A significantly higher expression level of the *eae* gene was observed in *eae*-ν strains (eg., ESA012, ESA138, and ESA177), with the transcript level ranging from 2.9 to 4.4-fold. However, no significant differences in *ler* transcript levels among strains with different *eae* subtypes or sources were found (Figure 3).

### 3.7. Adherence Patterns of E. albertii Isolates

In this study, 28 *E. albertii* strains belonging to 13 *eae* subtypes were investigated for their adherence to cultivated HEp-2 cells, and strain NBRC 107761 was used as a reference strain. All 28 *E. albertii* strains were different from the tEPEC strain E2348/69, which showed LA in 3 h-assay. Twelve strains (42.9%) displayed a LAL pattern and ten strains (35.7%) displayed a DA pattern, while six (21.4%) strains resulted in cell detachment (DE) in 6 h-assay (Table 2). Among six strains with DE patterns, five strains carried *cdtB*-VI and one carried *cdtB*-II. Human-derived strains exhibited LAL (6/8, 75%) or DA (2/8, 25%) patterns, while animal and food-derived strains exhibited three patterns. Additionally, none of the gene transcripts (*ler* or *eae*) showed significant differences among strains with LAL, DA, and DE patterns (Table 2 and Figure 3).

## 4. Discussion

Intimin, encoded by *eae* gene, plays a crucial role in the development of A/E lesions by inducing the effacement of microvilli and forming actin pedestals [15]. Currently, at least 30 *eae* subtypes have been defined in *E. coli* [15]. Previous studies indicated that *eae* subtypes are correlated with host specificity and tissue tropism in *E. coli* [15,32]. For example, the *eae*-γ1 appeared to be the most frequent subtype among O157:H7 strains isolated from patients with bloody diarrhea (BD) and hemolytic uremic syndrome (HUS) [15]. In atypical EPEC strains, *eae*-β1 was found to be significantly prevalent in strains derived from diarrheal patients, while eae-ζ3 was commonly observed in strains derived from cattle [32]. The correlation between different *eae* subtypes and disease severity or hosts could be considered in the risk assessment of bacterial infections.

Increasing collection of *E. albertii* from various sources has resulted in a rapid increase in the number of *eae* subtypes. Some intimin subtypes identified in *E. albertii* strains have been novel or rare subtypes in *E. coli*. In this study, *E. albertii* from diverse sources exhibited diversity in the *eae* subtypes. Based on the sequence polymorphism, a total of 23 *eae* subtypes were identified among 459 *E. albertii* strains. Two novel subtypes, *eae*-α11 and η3, were named. Moreover, the distribution of *eae* subtypes varied among human, animal, and food-derived strains. For instance, *eae*-γ, τ, and α11 subtypes were found exclusively in human-derived strains. *eae*-α9, ε4, and σ2 were predominantly present in animal-derived strains, while *eae*-ρ and σ were primarily found in food-derived strains. Furthermore, each *eae* subtype possessed different genotypes. ρ/GT4 and α9/GT6 were associated with animal-derived strains that carried *stx2f* genes, which was relevant to clinical disease in finches and mild symptoms in humans [7,33].

The LEE island provides the genetic basis of observed A/E lesions [34]. It was initially identified in EPEC, which was a major cause of diarrhea in infants and young children worldwide [35]. Subsequently, it was found to occur in EHEC, rabbit diarrheagenic *E. coli* (RDEC), the murine pathogen *Citrobacter rodentium*, and later in *E. albertii* [36]. The regulation of LEE island has been intensively investigated in A/E pathogens, including in *E. albertii* type strain Albert 19982^T^ (=NBRC 107761) [2]. In this study, the regulatory landscapes of diverse *E. albertii* strains were explored. The LEE elements in *E. albertii* strains were highly conserved, and their evolution was not synchronized with the genomic evolution. These findings were consistent with previous studies [34,37]. Moreover, the LEE elements in different strains exhibited diverse transcription and adherence patterns, which suggested their involvement in the pathogenic process. No significant differences in *ler* transcript levels among strains with different *eae* subtypes or sources were found, while a significantly higher expression level of the *eae* gene was observed in *eae*-ν strains. LAL was the most frequent adherence pattern among *E. albertii* strains, whereas DA and DE patterns were found in lower frequencies. Human-derived strains were more likely to exhibit LAL patterns, which might be relevant to the pathogenic process of colonization [38,39]. There might be other factors that could influence the transcription and expression of LEE elements, such as specific genes, plasmids, and other mobile elements. For instance, the LA pattern of tEPEC was highly related to the bundle-forming pilus encoded by EPEC adherence factor (EAF) plasmid. Moreover, some strains caused cell detachment which might be related to the subtype of *cdtB*-VI or other toxins [5]. Nevertheless, further studies were required to understand the in vivo pathogenicity of strain-specific *eae* variants, LEE transcription, and cell adherence.

In conclusion, we described the genetic diversity of *eae* gene in *E. albertii* strains isolated from different sources and identified two novel *eae* subtypes. Most *eae* subtypes were distributed among human, animal, and food-derived strains, while some subtypes showed host preference. The sequence and organization of LEE island among *E. albertii* isolates were relatively conserved, but the expression of *ler* and *eae* genes in different isolates was variable. Additionally, the LAL pattern represented a virulence property of *E. albertii* strains, especially human-derived strains. However, many strains exhibit DA or DE patterns. Further in vivo and in vitro studies are underway to understand the role of LEE in different genetic backgrounds.

## Figures and Tables

**Figure 1 microorganisms-11-02843-f001:**
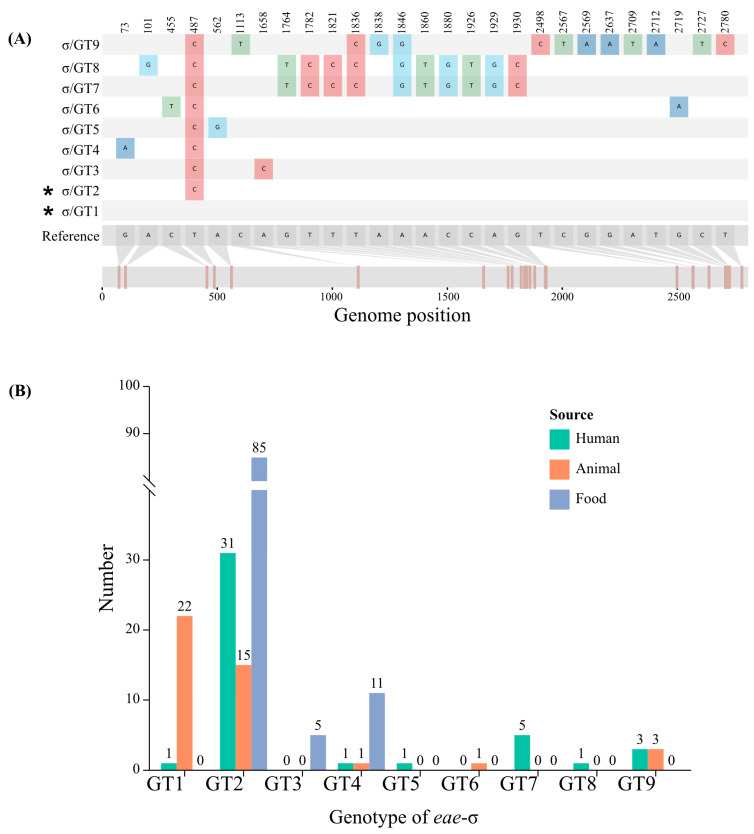
(**A**) The nucleotide differences among *eae*-σ GTs. The asterisk represented the synonymous mutation. (**B**) The distribution of *eae*-σ GTs among strains isolated from different sources.

**Figure 2 microorganisms-11-02843-f002:**
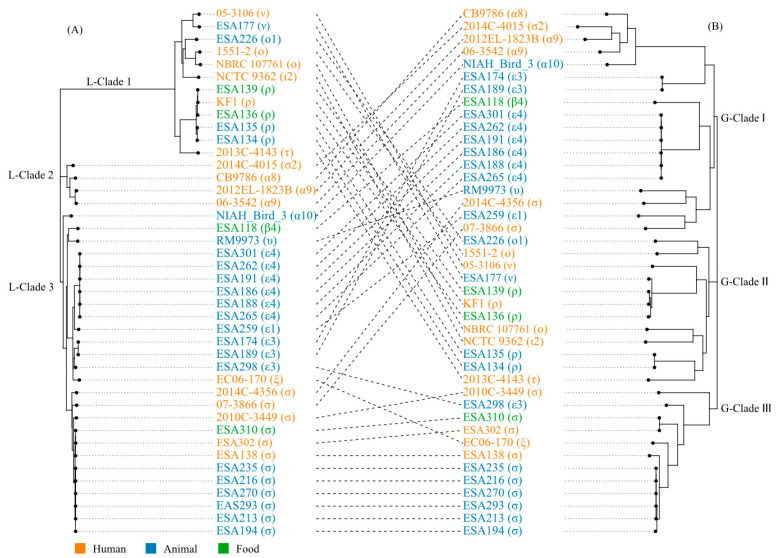
Tanglegram comparison between a phylogenetic tree based on sequences of LEE elements and pan-genome tree. (**A**) Neighbor-joining phylogenetic tree based on the sequences of the LEE elements. The tree was divided into three main clusters (L-Clades 1, 2, and 3). (**B**) A pan-genome tree is based on the presence or absence of the gene in the pan-genome. The tree is also divided into three clusters (G-Clades I, II, and III) according to topological structure and evolutionary distance. The colors of the leaves indicated the source of strains. Lines between trees link the same strains and crossing lines indicate a lack of similarity in the two trees.

**Figure 3 microorganisms-11-02843-f003:**
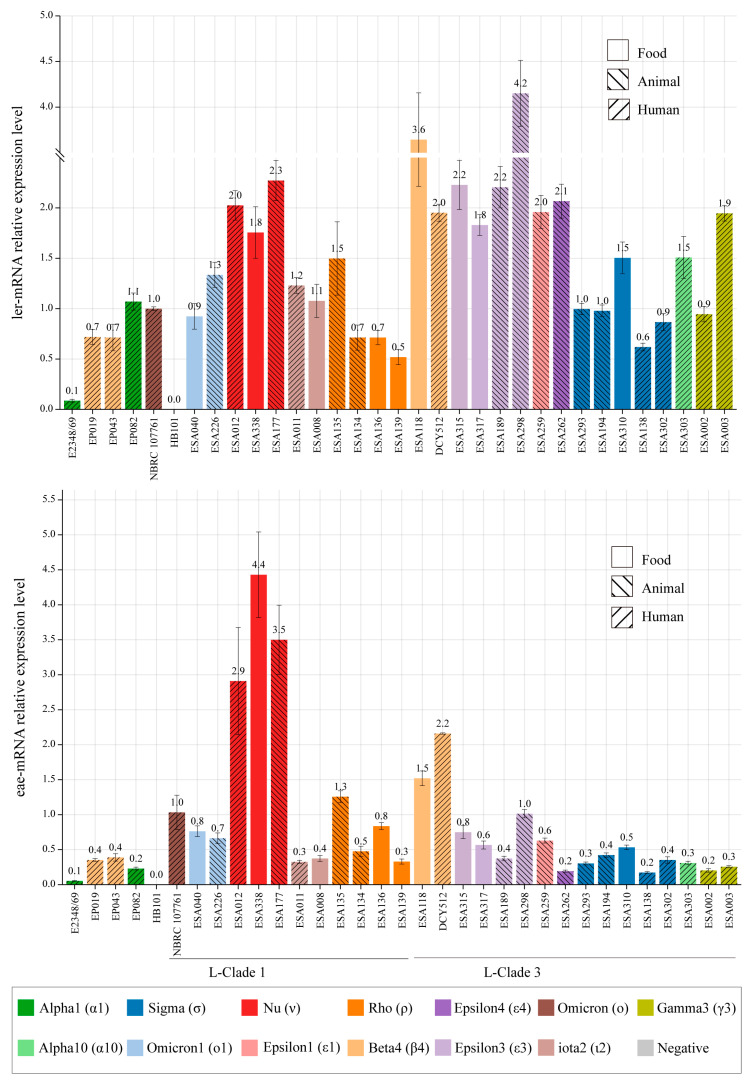
Relative transcription of *ler* and *eae* genes as determined by qRT-PCR.

**Table 1 microorganisms-11-02843-t001:** Intimin subtypes of *E. albertii* strains isolated from humans, animals, and food.

Subtypes	Human	Animal	Food	Total	*p* Value
alpha10 (α10)	10	8	0	18	0.001 *
alpha8 (α8)	6	4	0	10	0.019 *
alpha9 (α9)	5	20	0	25	<0.001 *
beta3 (β3)	3	8	0	11	0.004 *
beta4 (β4)	6	3	3	12	0.692
epsilon1 (ε1)	1	4	0	5	0.064
epsilon3 (ε3)	5	9	15	29	0.113
epsilon4 (ε4)	1	12	0	13	<0.001 *
gamma3 (γ3)	3	0	0	3	0.031 *
iota2 (ι2)	15	1	2	18	<0.001 *
lambda2 (λ2)	4	2	0	6	0.069
lambda3 (λ3)	1	0	0	1	0.317
nu (ν)	5	2	1	8	0.160
omicron (ο)	7	4	0	11	0.008 *
omicron1 (ο1)	1	2	4	7	0.574
rho (ρ)	9	2	36	47	<0.001 *
sigma (σ)	43	42	101	186	<0.001 *
sigma2 (σ2)	4	16	0	20	<0.001 *
tau (τ)	6	0	0	6	0.001 *
xi (ξ)	4	3	0	7	0.077
ypsilon (υ)	1	2	0	3	0.312
alpha11 (α11)	4	0	0	4	0.01 *
eta3 (η3)	0	3	0	3	0.067
Negative	3	3	0	6	0.077
Total	147	150	162	459	-

* Statistically significant difference.

**Table 2 microorganisms-11-02843-t002:** HEp-2 adherence assays of different *E. albertii* strains.

Strain Name	Sources	Detail of Sources	*eae* Subtypes	*cdtB* Subtypes	Cell Adherence
E2348/69	Human	Feces	α1	-	LA_3h_
EAEC 042	Human	Feces	-	-	AA_3h_
aEPEC EP019	Human	Feces	β4	-	LAL
HB101	Lab	Lab	-	-	NA
NBRC 107761	Human	Feces	ο	*cdtB*-VI	LAL
ESA040	Food	Duck intestine	ο1	*cdtB*-VI	DA
ESA226	Animal	TBG	ο1	*cdtB*-VI	DA
ESA012	Human	Feces	ν	*cdtB*-II	DA
ESA338	Food	Swine meat	ν	*cdtB*-VI	DA
ESA177	Animal	LWG	ν	*cdtB*-VI	DA
ESA011	Human	Feces	ι2	*cdtB*-II	DA
ESA008	Food	Duck intestine	ι2	*cdtB*-VI	LAL
ESA135	Animal	Bat	ρ	*cdtB*-VI	DE
ESA134	Animal	Bat	ρ	*cdtB*-VI	DE
ESA136	Food	Chicken intestine	ρ	*cdtB*-VI	DE
ESA139	Food	Duck intestine	ρ	*cdtB*-VI	DE
ESA118	Food	Duck intestine	β4	*cdtB*-II	LAL
DCY512	Human	Feces	β4	*cdtB*-I/II	LAL
ESA315	Food	Chicken intestine	ε3	*cdtB*-II	DA
ESA317	Food	Chicken meat	ε3	*cdtB*-II	LAL
ESA189	Animal	EW	ε3	*cdtB*-II	DE
ESA298	Animal	EW	ε3	*cdtB*-II	LAL
ESA259	Animal	EW	ε1	*cdtB*-II	LAL
ESA262	Animal	NP	ε4	*cdtB*-VI	DE
ESA293	Animal	EW	σ	*cdtB*-II	DA
ESA194	Animal	TBG	σ	*cdtB*-II	DA
ESA310	Food	Chicken intestine	σ	*cdtB*-II	LAL
ESA138	Human	Feces	σ	*cdtB*-II	LAL
ESA302	Human	Feces	σ	*cdtB*-II	LAL
ESA303	Human	Bloodstream	α10	*cdtB*-I/II	LAL
ESA002	Human	Feces	γ3	*cdtB*-VI	DA
ESA003	Human	Feces	γ3	*cdtB*-VI	LAL

‘-’, absent; LWG, Lesser white-fronted goose (*Anser erythropus*); TBG, Taiga bean goose (*Anser fabalis*); EW, Eurasian wigeon (*Mareca penelope*); GWG, Greater white-fronted goose (*Anser albifrons*); NP, Northern Pintail (*Anas acuta*); LAL: localized-like adherence; NA: nonadherent; AA: aggregative adherence; DA: diffuse adherence; LA: localized adherence; DE, detachment.

## Data Availability

The genomes of strain ESA302 were submitted to GenBank under the accession numbers GCA_032680885.1.

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
