# Peer review of "Genetic Diversity and Expression of Intimin in Escherichia albertii Isolated from Humans, Animals, and Food"

_microorganisms, 2023, doi:10.3390/microorganisms11122843_

Round 1
Reviewer 1 Report
Comments and Suggestions for Authors
The authors analyzed the LEE in a collection of E. albertii strains. They investigated the diversity of eae subtypes and found a large diversity of eae subtypes. Interestingly, while some eae subtypes were found exclusively in human-derived strains, some subtypes were mostly found in animal and food-derived strains.
Major comments:
The authors explored the transcript levels of ler and eae according to the eae subtype but couldn’t find any differences/association. A negative result is interesting but the authors fail to suggest any other explanation, venue to explore to explain the different eae expression levels in some of the strains.
The authors also explored the various adherence patterns of E. albertii isolates, but couldn’t find any association with strain source or ler and eae transcripts levels. In this particular case, because the WGS data are available the authors could have performed a GWAS to identify or exclude the presence of an additional factor that could explain the various adherence patterns.
Minor comments :
L107 : Please indicate the culture conditions of the strains before mRNA extraction: medium used, temperature, incubation time and conditions.
L120: Same remark. Please indicate culture conditions for the bacterial cultures used for HEp-2 infections.
L229 and L299: the authors are not clear concerning the concordance between LEE and pan-genome phylogeny. L229 they state that the results show a significant codivergence relationship (between) the LEE (and the pangenome), hence implying a degree of co-evolution. While L299 they state that the LEE evolution was not synchronized with the genomic evolution. This needs to be clarified in both sections.
Reviewer 2 Report
Comments and Suggestions for Authors
The manusctipt entitled Genetic diversity and expression of intimin in Escherichia albertii isolated from humans, animals, and food addresses an important issue related to the occurrence of the pathogenic species .
The work includes both bioinformatic analyses and experiments on a pool of selected isolates (mRNA expression level analyses of LEE-related genes and cell adherence assays). The materials and methods section does not make it clear enough that the experimental work was carried out on a selected group of isolates. Moreover, the authors declare (lines 68/69): one newly sequenced strain ESA302 in this study - but there is no information on the analysis methodology. In my opinion:
- in section 2.6 it should be clearly stated that mRNA was isolated from selected bacterial isolates
- in section 2.7 it should be clearly stated that analyses were performed on selected bacterial isolates
The article should be accepted after minor revision (corrections to minor methodological errors and text editing).
